# Design and Evaluation of a Geometric Algebra-Based Graph Neural Network for Molecular Property Prediction

Kasper Helverskov Petersen[*1] and Mikkel N. Schmidt[1]

[1]Technical University of Denmark, Kongens Lyngby, Denmark
{s203294, mnsc}@dtu.dk

## Abstract

Geometric Algebra (GA) provides a unified framework for representing scalars, vectors, and higher-dimensional geometric elements, along with the geometric product, an operation that mixes information across these components in an equivariant manner. While GA has recently attracted attention in deep learning, its potential for molecular property prediction remains underexplored. We introduce GA-GNN, a novel equivariant graph neural network that extends message passing architectures from separate scalar and vector features to multivector representations, and employs sequences of geometric product layers as the core update mechanism. Evaluated on the QM9 benchmark, GA-GNN achieves competitive performance with the recent state-of-the-art while demonstrating that GA-based representations can simplify architecture design. These results highlight the potential of GA for building expressive equivariant message passing networks for molecular property prediction.

## 1 Introduction

Equivariant neural networks have emerged as powerful tools for learning from data with geometric structure, such as molecules, by ensuring that learned features transform consistently under translation, rotation and reflection in 3-dimensional space. Current approaches often represent features as scalars, vectors, or higher-order tensors, with message passing architectures designed to respect these symmetries. We propose an alternative based on Geometric Algebra (GA), a unified mathematical framework for representing and manipulating geometric entities. GA extends beyond scalars and vectors to include geometric objects of higher dimensionality such as oriented planes and volumes, all combined in a single object called a multivector. A central operation in GA, the geometric product, mixes information not only within the same representational level (e.g., vector–vector) but also across levels (e.g., scalar–vector, vector–plane) in a principled manner that is equivariant to rotation and reflection. This makes GA a natural candidate for message passing architectures, where information from different geometric orders must be combined efficiently and consistently.

Through multivector representations and the geometric product, the network can propagate not only distances and directions but also learned notions of orientation, area, and volume, allowing it to express geometric relationships that standard GNNs cannot explicitly represent, while offering an alternative to the tensor and spherical-harmonic parameterizations used in current state-of-the-art equivariant models.

Though GA has recently attracted interest in deep learning, its potential for molecular property prediction remains underexplored. In this work, we present GA-GNN, a novel equivariant graph neural network for molecular property prediction. The architecture of the model is inspired by elements from PaiNN [1], which we extend to work on multivectors, rather than decoupled scalar and vector features. Its core is based on designing a new update block which uses sequences of geometric product layers as proposed in Clifford Group Equivariant Neural Networks (CGENNs) [2] to compute residual updates. Additionally, the readout layer can be simplified by removing target-specific networks, instead allowing flexible selection of the multivector components relevant to a given target property. We evaluate GA-GNN on the QM9 dataset and study several ablations and architectural variations. To the best of our knowledge, this is the first application of a GA-based model to molecular property prediction on QM9, offering new insights into the use of multivector representations and geometric product operations in this domain. An extended discussion is provided in [3].

## 2 Background & Related Work

### 2.1 Molecular Property Prediction

In molecular property prediction, the goal is to learn a function that maps molecular structures to their corresponding properties, which may include chemical, physical, or biological characteristics. In our setting, molecules are represented as graphs embedded in 3D space, where nodes correspond to atoms and edges capture chemical bonds or spatial proximity. Formally, a molecular graph is denoted $G = (V, E)$, where $V$ is the set of atoms (nodes) and $E \subseteq V \times V$ is the set of edges. Each atom $v \in V$ is associated with a spatial position $\mathbf{x}_v \in \mathbb{R}^3$ and an atom type, and each edge $(v, u) \in E$ may be asso-

---

*Corresponding Author.

Proceedings of the 7th Northern Lights Deep Learning Conference (NLDL), PMLR 307, 2026.

ciated with geometric features such as interatomic distance $\|\mathbf{x}_u - \mathbf{x}_v\|$ and relative position (edge vector) $\mathbf{r}_{vu} = \mathbf{x}_u - \mathbf{x}_v$. The neighborhood of a node $v$ is defined as $\mathcal{N}(v) = \{u \in V \mid (v, u) \in E\}$. These representations and features are used as inputs to graph-based deep learning models.

A wide range of deep learning methods have been developed for molecular property prediction. The majority of these approaches are based on message passing neural networks (MPNN). MPNNs learn node embeddings by iteratively aggregating and updating information from neighboring nodes. At each message passing round $t \in \{1, \ldots, T\}$, the embedding of node $v$ is denoted $\mathbf{h}_v^{(t)} \in \mathbb{R}^d$, where $d$ is the feature dimension. Each round consists of a message aggregation step followed by an update, and after $T$ rounds, a readout layer computes the final graph-level output based on the node embeddings:

$$\mathbf{m}_v^{(t+1)} = \bigoplus_{u \in \mathcal{N}(v)} M_t(\mathbf{h}_v^{(t)}, \mathbf{h}_u^{(t)}), \quad (1)$$

$$\mathbf{h}_v^{(t+1)} = U_t(\mathbf{h}_v^{(t)}, \mathbf{m}_v^{(t+1)}), \quad (2)$$

$$\hat{y} = R(\{\mathbf{h}_v^T \mid v \in V\}). \quad (3)$$

Here $M_t$ is called the *message function*, $U_t$ is called the *update function*, $\bigoplus$ is a permutation invariant aggregation operation (typically the sum), and $R$ is called the *readout function* [4].

Early MPNN-based models for molecular property prediction primarily relied on invariant, scalar-valued features such as pairwise interatomic distances, bond angles, and torsion angles [4–8]. More recent *equivariant* MPNNs incorporate vector-valued features that transform consistently under geometric transformations such as rotations and reflections [1, 9–11]. Later generations of models further extends this by using higher-order tensor features, which are updated through operations involving spherical harmonics and tensor products [12–15]. Non message-passing approaches including transformer models have also achieved promising results [16–20]. Lastly, other approaches that avoid using 3D geometric information instead operate on 2D molecular graphs, for example by learning motif-level representations, training topology-specific experts, or inducing a hierarchical grammar that defines a geometry over molecular graphs [21–23].

## 2.2 Geometric Algebra

**Definition.** Let $\{e_1, \ldots, e_n\}$ be the basis of an $n$-dimensional vector space $V$. The *geometric algebra* $\mathbb{G}_{pqr}$ is an algebra generated from the basis vectors $e_i$ in which the following two conditions hold:

1. For all $i$, the squared basis vectors satisfy:

$$e_i^2 = \begin{cases} +1 & \text{for } i = 1, \ldots, p, \\ -1 & \text{for } i = p+1, \ldots, p+q, \\ 0 & \text{for } i = p+q+1, \ldots, p+q+r, \end{cases} \quad (4)$$

**Table 1.** Basis blades in $\mathbb{G}_{3,0,0}$ grouped by grade.

| Name | Grade | Dimension | Basis blades | Square |
|------|-------|-----------|--------------|--------|
| Scalar | 0 | 1 | 1 | +1 |
| Vector | 1 | 3 | $e_1,\ e_2,\ e_3$ | +1 |
| Bivector | 2 | 3 | $e_{12},\ e_{23},\ e_{31}$ | −1 |
| Trivector | 3 | 1 | $e_{123}$ | −1 |

where the integers $p, q, r \geq 0$ count the basis vectors squaring to $+1$, $-1$ and $0$ respectively.

2. For $i \neq j$ the basis vectors anti-commute:

$$e_i e_j = -e_j e_i. \quad (5)$$

The total dimension of the space is $n = p + q + r$, and $\mathbb{G}_{pqr}$ has $2^n$ basis elements. In the remainder of this section, we focus on the specific case $\mathbb{G}_{3,0,0}$, which is the algebra used in the proposed GA-based GNN architecture. The *geometric product* of two vectors $\mathbf{a}$ and $\mathbf{b} \in V$ is defined as

$$\mathbf{ab} = \mathbf{a} \cdot \mathbf{b} + \mathbf{a} \wedge \mathbf{b}, \quad (6)$$

where $\mathbf{a} \cdot \mathbf{b}$ is the inner product known from traditional vector algebra, and $\mathbf{a} \wedge \mathbf{b}$ is the *outer product*, which is also called the wedge product. The outer product produces an object that represents an oriented plane spanned by $\mathbf{a}$ and $\mathbf{b}$, which is called a *bivector*.

More generally, the geometric product of $k$ basis vectors produces a *k-blade*. Since the basis vectors are orthogonal, the inner product between them equals zero. As a result, the geometric product between the basis vector reduces to the outer product. Using the shorthand convention $e_{ij} := e_i e_j$, we can form the following blades from the basis vectors:

$$e_{12} := e_1 e_2 = e_1 \wedge e_2,$$
$$e_{23} := e_2 e_3 = e_2 \wedge e_3,$$
$$e_{31} := e_3 e_1 = e_3 \wedge e_1,$$
$$e_{123} := e_1 e_2 e_3 = e_1 \wedge e_2 \wedge e_3. \quad (7)$$

The last blade $e_{123}$ is called a *trivector*. In $\mathbb{G}_{(3,0,0)}$ it is also referred to as the *pseudoscalar* as it is one-dimensional and changes sign under reflection. Geometrically it represents an oriented volume that encodes the handedness of space, meaning that the sign indicates whether the orientation is right-handed or left-handed, and the magnitude corresponds to the volume. Each blade has a *grade* equal to the dimension of the subspace it represents, i.e. grade-0 blades are scalars, grade-1 blades are vectors, grade-2 blades are bivectors and grade-3 vectors are trivectors. Table 1 summarizes the geometric algebra $\mathbb{G}_{3,0,0}$.

Linear combinations of blades of different grades are called *multivectors*. In $\mathbb{G}_{3,0,0}$ a multivector $A$ can be written as:

$$A = \underbrace{\lambda_0}_{\text{Scalar}} + \underbrace{\lambda_1 e_1 + \lambda_2 e_2 + \lambda_3 e_3}_{\text{Vector}}$$
$$+ \underbrace{\lambda_4 e_{12} + \lambda_5 e_{23} + \lambda_6 e_{31}}_{\text{Bivector}} + \underbrace{\lambda_7 e_{123}}_{\text{Trivector}}. \quad (8)$$

Hence, $A$ is an 8-dimensional object consisting of scalar, vector, bivector, and trivector components. The $k$-grade of a multivector is denoted $\langle A \rangle_k$. The geometric product is defined between two multivectors resulting in a new multivector that combines contributions from interactions between the grade-components of each multivector. The geometric product between multivectors in $\mathbb{G}_{(3,0,0)}$ is derived in Appendix A. This enables an organized way to mix information across representational levels of different dimensions all within a single consistent operation. Additionally, the geometric product is an equivariant operation under $O(3)$: Formally, for any orthogonal transformation $g \in O(3)$ and multivectors $A, B \in \mathbb{G}_{3,0,0}$ we have

$$(gA)(gB) = g(AB). \quad (9)$$

**Clifford Group Equivariant Neural Networks (CGENNs).** Recently several models based on combining GA with deep learning have been proposed, of which many are based on CGENNs [2]. CGENNs represent neurons in neural networks as multivectors and consist of *linear layers* and *geometric product layers* that operate on the multivector representations.

The linear layers operate independently on each grade of the multivectors using separate learnable transformations, and bias terms are included for the scalar components only:

$$\langle Y_i \rangle_k = \sum_j w_{ij}^{(k)} \langle X_j \rangle_k + \begin{cases} b_i, & k = 0, \\ 0, & \text{otherwise.} \end{cases} \quad (10)$$

Here $X_j$ and $Y_i$ are input and output multivectors respectively, and $w_{ij}^{(k)}$ and $b_i$ are learnable weights and biases. In this work we use a simplified linear layer where weights are shared across all blades, $w_{ij}^{(k)} = w_{ij}$. While both approaches preserve $O(3)$ equivariance, the former allows for greater per-grade expressiveness while the latter reduces the number of learnable parameters and emphasizes the idea of treating the multivector as a unified object rather than a collection of separate grades. In addition a bias term can be included for the trivector component; however, this breaks equivariance with respect to reflection.

The geometric product layers take the geometric product between pairs of multivectors and apply separate learnable weights for a total of 20 weights applied to a combination of 64 interaction pairs. Appendix B shows the derivation of the weighted geometric product between multivectors in $\mathbb{G}_{(3,0,0)}$.

Additionally, we refer to [2] for the original derivation of CGENN layers.

Recent work has explored the use of CGENN layers in message passing architectures on graphs by redesigning existing architectures such as EGNN, and testing on $n$-body simulation and protein denoising tasks [24]. These architectures maintain separate scalar and multivector embeddings, relying primarily on expressive scalar networks that occasionally interact with multivectors. In contrast, GA-GNN represents node states solely as multivectors and aggregates grade-wise messages to propagate geometric information during message passing. Furthermore, GA-GNN employs weighted geometric product layers with shared linear layers, whereas the methods in [24] use unparameterized geometric products and grade-wise linear layers alongside separate scalar networks. Finally, GA-GNN incorporates components such as continuous-filter convolutions and atom-type specific MLPs designed for molecular property prediction. Additional work has used CGENNs to perform message passing on simplicial complexes [25], construct Clifford-steerable kernels for convolutional neural networks [26], and design models for 3D molecular generation [27], and protein structure prediction [28].

## 3 Method

### 3.1 Architecture

Our architecture builds on the general framework of message-passing neural networks for molecules, taking PaiNN as a starting point of inspiration. Unlike PaiNN, which employs separate scalar and vector channels, we represent node states as multivectors, providing a unified representation across multiple geometric grades. The update block further introduces a novel update scheme based on successive geometric product layers, adapted from CGENN, to compute residual updates.

**Initialization.** Given an input graph $G = (V, E)$, we initialize each node $i$ with $F$ multivector channels in $\mathbb{G}_{(3,0,0)}$, which we denote:

$$A_i = \mathbf{s}_i + \vec{\mathbf{v}}_i + \vec{\mathbf{b}}_i + \mathbf{t}_i \in \mathbb{R}^{F \times 8}. \quad (11)$$

The four terms correspond to the scalar, vector, bivector, and trivector grades, respectively. Learned embeddings $\mathbf{u}_{z_i}^{(s)}, \mathbf{u}_{z_i}^{(t)} \in \mathbb{R}^F$ of the atom type $z_i$ associated with the node are used to initialize the scalar and trivector components, while vector and bivector components are initialized as zero:

$$\begin{aligned} \mathbf{s}_i^0 &= \mathbf{u}_{z_i}^{(s)} &\in \mathbb{R}^F, \\ \vec{\mathbf{v}}_i^0 &= \vec{\mathbf{0}} &\in \mathbb{R}^{F \times 3}, \\ \vec{\mathbf{b}}_i^0 &= \vec{\mathbf{0}} &\in \mathbb{R}^{F \times 3}, \\ \mathbf{t}_i^0 &= \mathbf{u}_{z_i}^{(t)} &\in \mathbb{R}^F. \end{aligned} \quad (12)$$

**Message block.** For each message passing round $t \in \{1, \ldots, T\}$, the message block computes and aggregates messages from sender nodes $j \in \mathcal{N}(i)$ to receiver nodes $i$. Figure 1 shows an overview of the message block architecture. Layer sizes (with feature dimension denoted $F$) are annotated in gray, and we denote elementwise multiplication by $\circ$. Similar to PaiNN, we apply continuous-filter convolutions from SchNet with an additional cosine cutoff function [29] to the pairwise edge distances:

$$\mathcal{W}(\|\mathbf{r}_{ij}\|) = f_{\text{cut}}(\|\mathbf{r}_{ij}\|) \cdot (\mathbf{W}_\psi \, \boldsymbol{\psi}(\|\mathbf{r}_{ij}\|) + \mathbf{b}_\psi) \,, \quad (13)$$

where $\psi$ denotes RBF expansion and $f_{cut}$ is the cosine cutoff function. The sender node's scalar state $\mathbf{s}_j$ is passed through a two-layer MLP $\phi(\mathbf{s}_j)$, and the transformed scalar features and edge features $\mathcal{W}_{ij}$ are combined via elementwise multiplication and split into 5 gates. One for each multivector component, and an additional gate for incorporating normalized edge vectors into the vector message:

$$\mathbf{g}_{ij} = \phi(\mathbf{s}_j) \circ \mathcal{W}_{ij} = \left[ \mathbf{g}_{ij}^{(s)}, \mathbf{g}_{ij}^{(v)}, \mathbf{g}_{ij}^{(d)}, \mathbf{g}_{ij}^{(b)}, \mathbf{g}_{ij}^{(t)} \right]. \quad (14)$$

Finally, the messages are computed by aggregating over neighbors in the following way:

$$\mathbf{m}_i^s = \sum_{j \in \mathcal{N}(i)} \mathbf{g}_{ij}^{(s)}, \quad (15)$$

$$\mathbf{m}_i^v = \sum_{j \in \mathcal{N}(i)} \mathbf{g}_{ij}^{(v)} \circ \vec{\mathbf{v}}_j + \mathbf{g}_{ij}^{(d)} \circ \frac{\mathbf{r}_{ij}}{\|\mathbf{r}_{ij}\|}, \quad (16)$$

$$\mathbf{m}_i^b = \sum_{j \in \mathcal{N}(i)} \mathbf{g}_{ij}^{(b)} \circ \vec{\mathbf{b}}_j, \quad (17)$$

$$\mathbf{m}_i^t = \sum_{j \in \mathcal{N}(i)} \mathbf{g}_{ij}^{(t)} \circ \mathbf{t}_j. \quad (18)$$

These messages are then added to the corresponding grades of the multivector state for the receiver nodes:

$$\begin{aligned} \mathbf{s}_i &\leftarrow \mathbf{s}_i + \mathbf{m}_i^s, \quad \vec{\mathbf{v}}_i \leftarrow \vec{\mathbf{v}}_i + \mathbf{m}_i^v, \\ \vec{\mathbf{b}}_i &\leftarrow \vec{\mathbf{b}}_i + \mathbf{m}_i^b, \quad \mathbf{t}_i \leftarrow \mathbf{t}_i + \mathbf{m}_i^t. \end{aligned} \quad (19)$$

Geometric product layers can also be added to the message block (see Appendix C), but this significantly increases computational cost, scaling with the number of edges rather than nodes, and our experiments with this indicate that the performance of this approach does not justify the overhead.

**Update block.** The update block processes each node's multivector representation using two linear projections followed by a sequence of geometric product layers and linear layers. Finally, the residual grade-wise update for each multivector is computed by summing over these transformations and modulated by grade-specific gates. Figure 2 shows an overview of the update block architecture. We denote weighted geometric product layers by $\text{GP}(A, B)_w$. We first compute two linear projections of the multivector state $A_i$: $U_i = \mathbf{U} \cdot A_i$ and

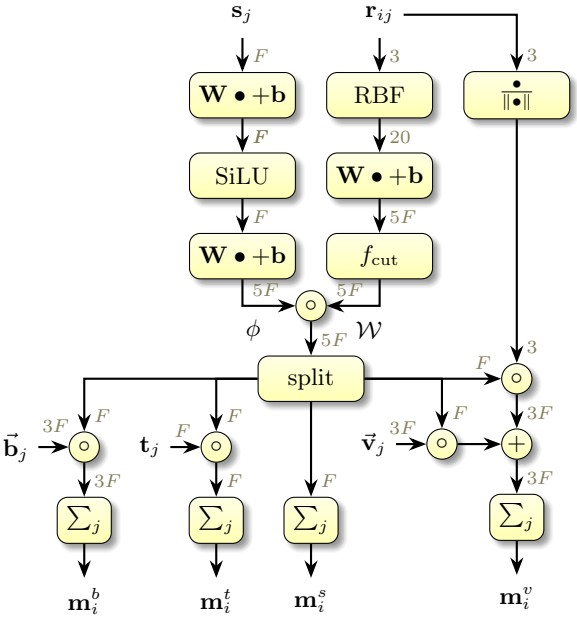

**Figure 1.** Overview of the message block architecture.

$V_i = \mathbf{V} \cdot A_i$. Then, we apply a sequence of weighted geometric products followed by linear layers:

$$Y_{n+1,i} = \mathbf{W}_{n+1} \cdot \text{GP}(X_{n,i}, Y_{n,i})_{w_{n+1}}, \quad (20)$$

for $n = 0, \ldots, N-1$ with $Y_{0,i} = V_i$, and where $X_{n,i}$ is either fixed as $U_i$, or set to $Y_{n-1,i}$ for $n \geq 2$ to create chained layers. In our main architecture, we use $N = 2$ and keep $X_{n,i} = U_i$, but the formulation supports chaining successive products by setting $X_{n,i} = Y_{n-1,i}$ for $n \geq 2$. To compute grade-specific gates for the residual update, we extract the scalar component from $A_i$ and compute the norm of the vector component from $V_i$. These are concatenated and passed through atom-type specific two-layer MLPs:

$$\mathbf{a}_i = \mathbf{W}_{2,z_i} \cdot \text{SiLU}(\mathbf{W}_{1,z_i}[\mathbf{s}_i, \|\langle V_i \rangle_1\|] + \mathbf{b}_{1,z_i}) + \mathbf{b}_{2,z_i}. \quad (21)$$

We split $\mathbf{a}_i \in \mathbb{R}^{F \times 4}$ into four separate gates, and compute residual updates for each grade of the multivector nodes:

$$\Delta \langle A_i \rangle_k = \mathbf{a}_i^{(k)} \circ \left( \langle U_i \rangle_k + \sum_{n=1}^N \langle Y_{n,i} \rangle_k \right) \quad (22)$$

Finally, the updated multivector representations for each node is given by adding the residual updates to each grade:

$$\langle A_i \rangle_k \leftarrow \langle A_i \rangle_k + \Delta \langle A_i \rangle_k \quad (23)$$

**Readout.** The readout layer maps multivector node states to graph-level predictions. We compare two approaches: (1) using PaiNN-style readout networks applied to specific multivector components,

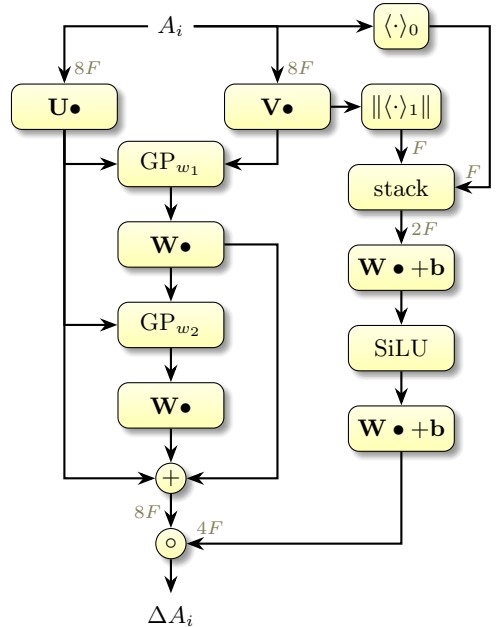

**Figure 2.** Overview of the update block architecture.

and (2) a simplified alternative where relevant components are summed directly across nodes and channels.

In the PaiNN-style setup, scalar properties are predicted by applying an MLP to the scalar part $\mathbf{s}_i$ of each node's multivector state:

$$f(\mathbf{s}_i) = \mathbf{W}_2 \cdot \text{SiLU}(\mathbf{W}_1 \mathbf{s}_i + \mathbf{b}_1) + \mathbf{b}_2, \qquad (24)$$

$$\hat{y}_G = \sum_{i \in G} f(\mathbf{s}_i). \qquad (25)$$

For the electronic spatial extent $\langle R^2 \rangle$, atom-wise contributions are weighted by squared distances:

$$\hat{y}_G^{\langle R^2 \rangle} = \sum_{i \in G} f(\mathbf{s}_i) \cdot \|\mathbf{x}_i\|^2. \qquad (26)$$

For the dipole moment $\mu$, the prediction is the magnitude of the vector:

$$\vec{\mu} = \sum_{i \in G} \vec{\mu}_{\text{atom}}(\vec{\mathbf{v}}_i) + q_{\text{atom}}(\mathbf{s}_i) \cdot \mathbf{x}_i, \qquad (27)$$

where $\vec{\mathbf{v}}_i = \langle A_i \rangle_1$ and $\mathbf{s}_i = \langle A_i \rangle_0$, and both $\vec{\mu}_{\text{atom}}$ and $q_{\text{atom}}$ are computed by summing over channels. In the base model, these components are passed through two gated equivariant blocks from PaiNN [1] with atom-type-specific MLPs beforehand.

The simplified readout layer, where we remove the output networks, sums directly across nodes and

channels for the relevant grade(s):

$$\hat{y}_G = \sum_{i \in G} \sum_{c=1}^{F} \mathbf{s}_{i,c}, \qquad (28)$$

$$\hat{y}_G^{\langle R^2 \rangle} = \sum_{i \in G} \left( \sum_{c=1}^{F} \mathbf{s}_{i,c} \right) \cdot \|\mathbf{x}_i\|^2, \qquad (29)$$

$$\vec{\mu} = \sum_{i \in G} \left( \sum_{c=1}^{F} \vec{\mathbf{v}}_{i,c} + \sum_{c=1}^{F} \mathbf{s}_{i,c} \cdot \mathbf{x}_i \right). \qquad (30)$$

This approach yields better performance on most targets in our ablation experiments and allows for greater architectural flexibility and generalization as it avoids specialized target-specific readout networks.

### 3.2 Dataset

We evaluate the proposed architecture on the QM9 dataset [30, 31]. The dataset consists of data for 130,381 small molecules, where atoms can be either carbon, hydrogen, oxygen, nitrogen or fluorine. We use the version of the dataset published by PyTorch Geometric [32, 33]. Preprocessing of the dataset consists of adding edges to the molecular graphs based on a cutoff distance between nodes in 3D space rather than using chemical bonds. We use $r_c = 5.0$ Å. The neighborhood for each node is thus given by:

$$\mathcal{N}(i) = \{ j \in \{1, \ldots, N\} \setminus \{i\} \mid \|\tilde{\mathbf{x}}_j - \tilde{\mathbf{x}}_i\| < r_c \}. \qquad (31)$$

Additionally, node coordinates are centralized according to atomic mass. Given a molecule consisting of $N$ atoms, each with atomic number $z_i$ and 3D position $\mathbf{x}_i \in \mathbb{R}^3$ for $i = 1, \ldots, N$, we first compute the center of mass using atomic masses $m_i$ for each atom type:

$$\mathbf{x}_{\text{com}} = \frac{1}{M} \sum_{i=1}^{N} m_i \mathbf{x}_i, \quad \text{where } M = \sum_{i=1}^{N} m_i. \quad (32)$$

All atom positions are then centralized by subtracting the center of mass: $\tilde{\mathbf{x}}_i = \mathbf{x}_i - \mathbf{x}_{\text{com}}$, and for each edge $(i, j)$ in the constructed graph, we compute an edge vector $\mathbf{r}_{ij} = \tilde{\mathbf{x}}_j - \tilde{\mathbf{x}}_i$.

Finally, for some QM9 targets, the dataset provides *atomic reference values* i.e., per-atom baseline contributions. For the targets that include atomrefs (targets 6-11), we subtract the sum of the corresponding atomic reference values from the target during training, so the model learns to predict only the residual. At test time, we add back the atomic reference contribution to obtain the final prediction.

### 3.3 Training details

We use the same hyperparameters as PaiNN [1]. All experiments use the AdamW optimizer [34] with

**Table 2.** MAE on 4 QM9 target properties for each addition/ablation compared to the base model. Bold results have the lowest error. Detailed descriptions of each experiment can be found in Appendix E.

| | $\epsilon_{\text{HOMO}}$ meV | $\mu$ D | $R^2$ $\alpha_0^2$ | $\alpha$ $\alpha_0^3$ |
|---|---|---|---|---|
| **Addition** | | | | |
| Sender/receiver GP | 26.5374 | 0.0127 | 0.1057 | 0.0525 |
| Sender/copy GP | 26.2839 | 0.0122 | 0.0843 | 0.0538 |
| 3 GP in update block | 24.5228 | 0.0127 | 0.0711 | 0.0525 |
| Grade-wise linear layers | 24.7583 | 0.0130 | **0.0661** | 0.0526 |
| **Ablation** | | | | |
| Removal of second GP | 23.4886 | NaN | 0.0868 | **0.0479** |
| Non-weighted GPs | 24.4641 | 0.0128 | 0.0806 | 0.0529 |
| No output networks | 28.0492 | **0.0119** | 0.0697 | 0.0506 |
| Trivectors initialized as 0 | **23.4371** | 0.0123 | 0.0878 | 0.0508 |
| Shared update MLP | 25.1560 | 0.0130 | 0.0772 | 0.0543 |
| Base architecture | 24.3626 | 0.0126 | 0.0731 | 0.0532 |

weight decay $\lambda = 0.01$. We use MSE as the loss function for all targets, except for $\alpha$ which uses MAE loss. If the validation loss plateaus, the learning rate is decayed by a factor of 0.5, with patience 5, and we use early stopping with patience 30. For learning rate decay and early stopping we use exponential smoothing of the validation loss with factor 0.9. When conducting the main evaluation, we found that some targets occasionally exhibit large validation-loss spikes early in training, which can prematurely trigger learning-rate decay or early stopping. To mitigate this, we discard validation-loss values that exceed twice the current smoothed validation loss for the eight targets not included in the architecture-selection study. Lastly, for the scalar properties that do not have atomic reference values, we normalize the target values before training and de-normalize when evaluating on the test set. The full set of hyperparameters are listed in Appendix D. The code for conducting the experiments can be found at https://github.com/khelverskovp/GA-GNN

# 4 Results

## 4.1 Architecture selection study

To explore the space of design choices, we first evaluated several architectural variations of the base model on four selected QM9 properties (Table 2). To limit the computational cost, these studies were carried out using only $F = 64$ channels. A detailed description of each variation and ablation can be found in Appendix E.

Results show that no single variant performs best across all targets, suggesting that different architectural choices benefit different molecular properties. However, the differences are generally modest, and since each variant was evaluated from a single training run, the results should not be over-interpreted.

Among the message block variants, both the geometric product between sender and receiver pairs and between sender nodes and linear projections of themselves perform worse than the base model on most targets. The latter does slightly improve $\mu$, but worsens all other targets. These results indicate that using the geometric product to mix between grades within a node to update its state is more effective than using it to combine features across neighboring nodes.

Adding a third geometric product in the update block shows mixed results, slightly improving performance on $\langle R^2 \rangle$ and $\alpha$, but slightly worsening $\epsilon_{\text{HOMO}}$ and $\mu$. This suggests diminishing returns from stacking additional geometric products in this setting. Conversely, while ablating the second GP layer improves $\epsilon_{\text{HOMO}}$ and $\alpha$, it leads to training instability on $\mu$, and worse performance on $\langle R^2 \rangle$.

Replacing the shared linear layers with grade-wise linear layers, while theoretically more expressive, leads to degraded performance on $\epsilon_{\text{HOMO}}$ and $\mu$, and comes at higher computational cost. For $\alpha$ it yields a slight improvement, and for $\langle R^2 \rangle$ it achieves the best results.

Removing the learnable weights from the geometric product layers leads to a consistent, though modest, degradation in accuracy. This suggests that most of the benefit comes from the structure of the geometric product itself, with the weights serving to refine the computation. Hence, in resource-constrained settings, the weights can possibly be omitted to reduce complexity, with only a small drop in performance.

Most notably, removing the gated equivariant blocks (for $\mu$ prediction) and readout MLP (for scalar targets and $\langle R^2 \rangle$) in the output layer improves performance on 3 out of 4 targets. It achieves the lowest error on $\mu$ and improves $\langle R^2 \rangle$ and $\alpha$, but does harm performance on $\epsilon_{\text{HOMO}}$. This suggests that multivectors may naturally encode enough information to predict certain molecular properties without additional processing.

Initializing trivectors as zero instead of using learned embeddings of the atom type leads to the best overall result for $\epsilon_{\text{HOMO}}$, and improves performance slightly on $\mu$ and $\alpha$, but worsens $\langle R^2 \rangle$.

Finally, using a shared update MLP across atom types harms performance across the board, confirming the value of atom-type-specific gates for the residual grade updates.

## 4.2 Main Evaluation

Based on these findings, we perform the main evaluation on all twelve QM9 properties. Table 3 presents a comparison between GA-GNN and state-of-the-art baseline models. Results for baselines are from [20, 37], and results for GA-GNN are averaged over 3 random data splits. We increased the feature dimension to $F = 128$, except for $\langle R^2 \rangle$, where $F = 64$ performed best. For $\epsilon_{\text{HOMO}}$ we initialize the trivector as zero instead of using atom type embeddings, and we only use one geometric product in the update block. For $\mu$ and $\langle R^2 \rangle$ we compute the final

**Table 3.** Average MAE of GA-GNN across three random splits, compared with state-of-the-art models, as reported in the literature, on QM9 targets. Models are ordered by their average rank across the targets. The lowest errors are shown in bold, and results within 10% of the best are underlined.

| | $\epsilon_{\text{HOMO}}$ meV | $\epsilon_{\text{LUMO}}$ meV | $\Delta\epsilon$ meV | $\mu$ D | $R^2$ $\alpha_0^2$ | $\alpha$ $\alpha_0^3$ | ZPVE meV | $U_0$ meV | $U$ meV | $H$ meV | $G$ meV | $c_v$ $\frac{\text{cal}}{\text{mol K}}$ | Avg. rank |
|---|---|---|---|---|---|---|---|---|---|---|---|---|---|
| **Cormorant** [12] | 34 | 38 | 61 | 0.038 | 0.961 | 0.085 | 2.03 | 22 | 21 | 21 | 20 | 0.026 | 16.92 |
| **LieConv** [9] | 30 | 25 | 49 | 0.032 | 0.800 | 0.084 | 2.28 | 19 | 19 | 24 | 22 | 0.038 | 16.42 |
| **NMP** [4] | 43 | 38 | 69 | 0.030 | 0.180 | 0.092 | 1.50 | 20 | 20 | 17 | 19 | 0.040 | 16.00 |
| **SchNet** [5] | 41 | 34 | 63 | 0.033 | 0.073 | 0.235 | 1.70 | 14 | 19 | 14 | 14 | 0.033 | 14.83 |
| **MGCN** [6] | 42 | 57 | 64 | 0.056 | 0.110 | **0.030** | 1.12 | 13 | 14 | 16 | 15 | 0.038 | 13.17 |
| **SEGNN** [13] | 24 | 21 | 42 | 0.023 | 0.660 | 0.060 | 1.62 | 15 | 13 | 13 | 15 | 0.031 | 13.17 |
| **EGNN** [10] | 29 | 25 | 48 | 0.029 | 0.106 | 0.071 | 1.55 | 11 | 12 | 12 | 12 | 0.031 | 12.58 |
| **EQGAT** [11] | 20 | 16 | 32 | 0.011 | 0.382 | 0.053 | 2.00 | 25 | 25 | 24 | 23 | 0.024 | 12.08 |
| **NoisyNodes** [35] | 20 | 19 | 29 | 0.025 | 0.700 | 0.052 | 1.16 | 7.3 | 7.6 | 7.4 | 8.3 | 0.025 | 9.67 |
| **DimeNet++** [7] | 25 | 20 | 33 | 0.030 | 0.331 | 0.044 | 1.21 | 6.3 | 6.3 | 6.5 | 7.6 | 0.023 | 8.42 |
| **SphereNet** [8] | 23 | 18 | 32 | 0.026 | 0.292 | 0.046 | 1.12 | 6.3 | 6.4 | 6.3 | 7.8 | 0.022 | 7.42 |
| **TorchMD-NET** [17] | 20 | 18 | 36 | 0.011 | 0.033 | 0.059 | 1.84 | 6.2 | 6.4 | 6.2 | 7.6 | 0.026 | 7.25 |
| **PaiNN** [1] | 28 | 20 | 46 | 0.012 | 0.066 | 0.045 | 1.28 | 5.9 | 5.8 | 6.0 | 7.4 | 0.024 | 7.08 |
| **Equiformer** [18] | 15 | 14 | 30 | 0.011 | 0.251 | 0.046 | 1.26 | 6.6 | 6.7 | 6.6 | 7.6 | 0.023 | 6.75 |
| **GNS-TAT+NN** [36] | 17 | 17 | 26 | 0.021 | 0.650 | 0.047 | **1.08** | 6.4 | 6.4 | 6.4 | 7.4 | 0.022 | 6.25 |
| **MACE** [14] | 22 | 19 | 42 | 0.015 | 0.210 | 0.038 | 1.23 | 4.1 | 4.1 | 4.7 | 5.5 | 0.021 | 5.67 |
| **Equiformer V2** [19] | **14** | **13** | 29 | 0.010 | 0.186 | 0.050 | 1.47 | 6.2 | 6.5 | 6.2 | 7.6 | 0.023 | 5.50 |
| **GotenNet B** [20] | 15 | **13** | **21** | **0.007** | **0.027** | 0.032 | 1.09 | **3.4** | **3.5** | **3.4** | **5.2** | **0.019** | 1.25 |
| **GA-GNN** | 21 | 18 | 36 | 0.011 | 0.063 | 0.045 | 1.18 | 6.2 | 6.2 | 6.1 | 7.2 | 0.023 | 5.00 |

predictions without output networks (using Eq. 30 and 29, respectively). For $\langle R^2 \rangle$, we additionally test both with and without grade-wise linear layers and find that the effect on performance is minimal (MAE of 0.063 vs. 0.064). For $\alpha$ we also only use one geometric product in the update block. For the eight scalar properties not included in the architecture selection study we use the same architecture as was used for $\epsilon_{\text{HOMO}}$ except for the cases of ZPVE, $U_0$ and $U$, where we additionally remove the output network in the readout layer (i.e. using Eq. 28).

The results show that GA-GNN is generally competitive across all properties. However, we observe strong performance on targets that have a clear geometric interpretation that aligns well with the representational capacity of the multivector structure. For example, the model achieves rank 3 on the dipole moment $\mu$ and electronic spatial extent $\langle R^2 \rangle$, which are both linked to spatial geometry: the dipole moment is a vector defined by the distribution of partial charges and their relative positions, while $\langle R^2 \rangle$ reflects the squared-distance spread of the electronic density. Meanwhile, the model performs relatively worse on the molecular orbital energy targets ($\epsilon_{\text{HOMO}}$, $\epsilon_{\text{LUMO}}$, and $\Delta\epsilon$), possibly because these quantities depend on aspects of molecular structure that are not strongly reflected in the geometric information that GA-GNN captures. Overall across the 12 targets, the model ranks fifth on average, surpassed only by GotenNet [20]. Results for each split are included in Appendix F.

## 5 Discussion

This work demonstrates that geometric algebra provides a viable and powerful framework for designing expressive message-passing networks for molecular property prediction. By representing nodes as mul-

tivectors and updating them through weighted geometric products, GA-GNN attains competitive performance with recent state-of-the-art models such as MACE and Equiformer-V2.

The architecture is flexible in the sense that multivectors contain scalars, vectors, bivectors and trivectors in a single structure, and the geometric product mixes information across these components in a consistent way. This means the model does not require separate architectural modules for different geometric quantities; instead the same representational framework can be applied to targets with different geometric characteristics requiring only selecting the relevant grades in the readout layer. That said, our architecture selection study shows that this unification does not imply that a single architectural configuration is universally optimal across all targets. Design choices such as the number of geometric product layers affect targets in different ways. Because the study is based on single training runs and the observed differences are modest, they should be interpreted with caution. Still, they indicate that while GA-GNN provides a unified representational framework, achieving optimal performance across targets remains sensitive to specific architectural decisions, consistent with general patterns observed in molecular property prediction.

### 5.1 Comparison to GotenNet

Although GA-GNN performs competitively, its primary contribution is to establish and explore geometric algebra as a viable architectural approach for molecular property prediction rather than to serve as a hyperparameter-optimized state-of-the-art benchmark. The performance gap relative to the recently proposed GotenNet model therefore still leaves open the question of when and under which conditions

GA-based representations should be preferred.

GotenNet encodes node and edge features using spherical harmonics up to degree 2. While the scalar and vector components of GA-GNN align conceptually with degree-0 and degree-1 spherical harmonics (capturing distance and direction), the bivector and trivector components do not correspond to degree-2 spherical harmonics. Degree-2 harmonics span a five-dimensional space of quadrupole-like angular patterns that describe second-order directional variation (e.g., elongation, planar spread, anisotropy). GotenNet's superior performance suggests that such degree-2 features may capture molecular interactions more effectively than the oriented planes and volumes represented by bi- and trivectors, respectively.

On the other hand, GA-GNN is competitive with other architectures that are based on spherical harmonics (e.g. MACE, Equiformer V2) indicating that the choice of representational formalism alone cannot explain the observed performance gap. Comparing the two models in terms of architectural design, a key difference is that GotenNet employs attention-based message passing which GA-GNN does not. While this is also the case in Equiformer V2, the specific attention design in GotenNet appears to be more expressive, using two complementary attention blocks to update node and scalar edge features respectively. These blocks compute interaction strengths jointly from both sender and receiver node features and the connecting edge, whereas GA-GNN's message weighting, based on continuous filter convolutions, depends only on the sender's features and the pair's distance. This attention based approach may provide a more flexible mechanism for gating information flow between nodes, thus better allowing GotenNet to determine which neighbors' information should dominate based on contextual relevance rather than the simpler approach used in GA-GNN.

If we compare the performance of transformer-based models in general (Equiformer, GNS-TAT+NN, and GotenNet) against non-transformers, including GA-GNN, we observe that these methods show superior performance on $\epsilon_{HOMO}, \epsilon_{LUMO}$, and $\Delta\epsilon$ in particular. Similarly, EQGAT, which introduces attention into a PaiNN-like architecture, achieves substantial improvements on these specific targets relative to its performance on the remaining targets. This suggests that these properties in particular benefit from attention-based message passing, which coincides with GA-GNN attaining its weakest relative performance on this subset.

Turning to computational considerations, GA-GNN incurs a higher computational cost and parameter count than earlier models such as PaiNN, largely due to the weighted geometric products and the atom-type–specific update MLPs. However, relative to recent state-of-the-art architectures, GA-GNN is not computationally demanding. Complexity analysis shows that it is comparable to GotenNet-B and has fewer parameters and a faster runtime than Equiformer-V2 (see Appendix G).

## 5.2 Future Work

Overall, our findings demonstrate both the feasibility and promise of GA-based GNNs for molecular property prediction, while also indicating several directions for further development. The factors discussed above may account for part of the performance gap to GotenNet, but the list is not exhaustive; additional architectural or representational elements may contribute as well, and identifying them would require a more systematic investigation.

Further directions for future work include exploring the generality of the framework by assessing its performance across additional datasets and domains. In line with the above discussion, experimenting with alternative message block designs that incorporate attention mechanisms may yield further improvements. Moreover, designing better initialization schemes for the multivector components—rather than initializing vector and bivector components as zero—may enhance the architecture, as may improving the simplified readout layers.

We have also not explored the use of multivector normalization layers from CGENN [2], or hybrid approaches such as maintaining separate node or edge features in addition to multivector features. Finally, exploring alternative GA spaces and conducting a systematic comparison may provide additional insights. For example $\mathbb{G}_{3,0,1}$ would expand the multivector representations from 8 to 16 components, which could provide additional expressive power and capture richer geometric structures. However it would increase the computational complexity of the framework.

## 6 Conclusion

We introduced GA-GNN, an equivariant graph neural network that extends the message-passing framework to multivector representations and employs geometric product layers from CGENN for structured feature interactions. Evaluated on the QM9 benchmark, GA-GNN achieves competitive performance with recent state-of-the-art models, demonstrating the feasibility and potential of GA-based representations for molecular property prediction. Our experiments highlight effective design choices for incorporating geometric product layers into message passing, as well as the use of shared linear layers. These findings open several directions for future work, including a broader evaluation of the approach and continued exploration through architectural refinements.

## Acknowledgments

This work was supported by the Novo Nordisk Foundation under grant no NNF22OC0076658 (Bayesian neural networks for molecular discovery) and by a grant from G-Research.

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

# Supplementary information

# A    Geometric product in $\mathbb{G}_{(3,0,0)}$

The geometric product is defined between all basis blades. Table A.1 from [38] shows the product between all pairs of basis blades in $\mathbb{G}_{3,0,0}$. Given two multivectors $A$ and $B$ in $\mathbb{G}_{(3,0,0)}$:

$$A = \lambda_0 + \lambda_1 e_1 + \lambda_2 e_2 + \lambda_3 e_3 + \lambda_4 e_{12} + \lambda_5 e_{23} + \lambda_6 e_{31} + \lambda_7 e_{123} \tag{33}$$
$$B = \beta_0 + \beta_1 e_1 + \beta_2 e_2 + \beta_3 e_3 + \beta_4 e_{12} + \beta_5 e_{23} + \beta_6 e_{31} + \beta_7 e_{123} \tag{34}$$

We can write the product $AB$ as:

$$
\begin{aligned}
AB = {} & \lambda_0(\beta_0 + \beta_1 e_1 + \beta_2 e_2 + \beta_3 e_3 + \beta_4 e_{12} + \beta_5 e_{23} + \beta_6 e_{31} + \beta_7 e_{123}) \\
& + \lambda_1 e_1(\beta_0 + \beta_1 e_1 + \beta_2 e_2 + \beta_3 e_3 + \beta_4 e_{12} + \beta_5 e_{23} + \beta_6 e_{31} + \beta_7 e_{123}) \\
& + \lambda_2 e_2(\beta_0 + \beta_1 e_1 + \beta_2 e_2 + \beta_3 e_3 + \beta_4 e_{12} + \beta_5 e_{23} + \beta_6 e_{31} + \beta_7 e_{123}) \\
& + \lambda_3 e_3(\beta_0 + \beta_1 e_1 + \beta_2 e_2 + \beta_3 e_3 + \beta_4 e_{12} + \beta_5 e_{23} + \beta_6 e_{31} + \beta_7 e_{123}) \\
& + \lambda_4 e_{12}(\beta_0 + \beta_1 e_1 + \beta_2 e_2 + \beta_3 e_3 + \beta_4 e_{12} + \beta_5 e_{23} + \beta_6 e_{31} + \beta_7 e_{123}) \\
& + \lambda_5 e_{23}(\beta_0 + \beta_1 e_1 + \beta_2 e_2 + \beta_3 e_3 + \beta_4 e_{12} + \beta_5 e_{23} + \beta_6 e_{31} + \beta_7 e_{123}) \\
& + \lambda_6 e_{31}(\beta_0 + \beta_1 e_1 + \beta_2 e_2 + \beta_3 e_3 + \beta_4 e_{12} + \beta_5 e_{23} + \beta_6 e_{31} + \beta_7 e_{123}) \\
& + \lambda_7 e_{123}(\beta_0 + \beta_1 e_1 + \beta_2 e_2 + \beta_3 e_3 + \beta_4 e_{12} + \beta_5 e_{23} + \beta_6 e_{31} + \beta_7 e_{123}) \tag{35}
\end{aligned}
$$

Expanding all products explicitly, using table A.1 gives:

$$
\begin{aligned}
AB = {} & \lambda_0\beta_0 + \lambda_0\beta_1 e_1 + \lambda_0\beta_2 e_2 + \lambda_0\beta_3 e_3 + \lambda_0\beta_4 e_{12} + \lambda_0\beta_5 e_{23} + \lambda_0\beta_6 e_{31} + \lambda_0\beta_7 e_{123} \\
& + \lambda_1\beta_0 e_1 + \lambda_1\beta_1 + \lambda_1\beta_2 e_{12} - \lambda_1\beta_3 e_{31} + \lambda_1\beta_4 e_2 + \lambda_1\beta_5 e_{123} - \lambda_1\beta_6 e_3 + \lambda_1\beta_7 e_{23} \\
& + \lambda_2\beta_0 e_2 - \lambda_2\beta_1 e_{12} + \lambda_2\beta_2 + \lambda_2\beta_3 e_{23} - \lambda_2\beta_4 e_1 + \lambda_2\beta_5 e_3 + \lambda_2\beta_6 e_{123} + \lambda_2\beta_7 e_{31} \\
& + \lambda_3\beta_0 e_3 + \lambda_3\beta_1 e_{31} - \lambda_3\beta_2 e_{23} + \lambda_3\beta_3 + \lambda_3\beta_4 e_{123} - \lambda_3\beta_5 e_2 + \lambda_3\beta_6 e_1 + \lambda_3\beta_7 e_{12} \\
& + \lambda_4\beta_0 e_{12} - \lambda_4\beta_1 e_2 + \lambda_4\beta_2 e_1 + \lambda_4\beta_3 e_{123} - \lambda_4\beta_4 - \lambda_4\beta_5 e_{31} + \lambda_4\beta_6 e_{23} - \lambda_4\beta_7 e_3 \\
& + \lambda_5\beta_0 e_{23} + \lambda_5\beta_1 e_{123} - \lambda_5\beta_2 e_3 + \lambda_5\beta_3 e_2 + \lambda_5\beta_4 e_{31} - \lambda_5\beta_5 - \lambda_5\beta_6 e_{12} - \lambda_5\beta_7 e_1 \\
& + \lambda_6\beta_0 e_{31} + \lambda_6\beta_1 e_3 + \lambda_6\beta_2 e_{123} - \lambda_6\beta_3 e_1 - \lambda_6\beta_4 e_{23} + \lambda_6\beta_5 e_{12} - \lambda_6\beta_6 - \lambda_6\beta_7 e_2 \\
& + \lambda_7\beta_0 e_{123} + \lambda_7\beta_1 e_{23} + \lambda_7\beta_2 e_{31} + \lambda_7\beta_3 e_{12} - \lambda_7\beta_4 e_3 - \lambda_7\beta_5 e_1 - \lambda_7\beta_6 e_2 - \lambda_7\beta_7 \tag{36}
\end{aligned}
$$

**Table A.1.**  Geometric product table for the basis blades in $\mathbb{G}_{3,0,0}$.

| GP | $\lambda$ | $e_1$ | $e_2$ | $e_3$ | $e_{12}$ | $e_{23}$ | $e_{31}$ | $e_{123}$ |
|---|---|---|---|---|---|---|---|---|
| $\lambda$ | $\lambda^2$ | $\lambda e_1$ | $\lambda e_2$ | $\lambda e_3$ | $\lambda e_{12}$ | $\lambda e_{23}$ | $\lambda e_{31}$ | $\lambda e_{123}$ |
| $e_1$ | $\lambda e_1$ | $1$ | $e_{12}$ | $-e_{31}$ | $e_2$ | $e_{123}$ | $-e_3$ | $e_{23}$ |
| $e_2$ | $\lambda e_2$ | $-e_{12}$ | $1$ | $e_{23}$ | $-e_1$ | $e_3$ | $e_{123}$ | $e_{31}$ |
| $e_3$ | $\lambda e_3$ | $e_{31}$ | $-e_{23}$ | $1$ | $e_{123}$ | $-e_2$ | $e_1$ | $e_{12}$ |
| $e_{12}$ | $\lambda e_{12}$ | $-e_2$ | $e_1$ | $e_{123}$ | $-1$ | $-e_{31}$ | $e_{23}$ | $-e_3$ |
| $e_{23}$ | $\lambda e_{23}$ | $e_{123}$ | $-e_3$ | $e_2$ | $e_{31}$ | $-1$ | $-e_{12}$ | $-e_1$ |
| $e_{31}$ | $\lambda e_{31}$ | $e_3$ | $e_{123}$ | $-e_1$ | $-e_{23}$ | $e_{12}$ | $-1$ | $-e_2$ |
| $e_{123}$ | $\lambda e_{123}$ | $e_{23}$ | $e_{31}$ | $e_{12}$ | $-e_3$ | $-e_1$ | $-e_2$ | $-1$ |

Finally, we collect like terms by grade:

$$
\begin{aligned}
AB = {} & \lambda_0\beta_0 + \lambda_1\beta_1 + \lambda_2\beta_2 + \lambda_3\beta_3 - \lambda_4\beta_4 - \lambda_5\beta_5 - \lambda_6\beta_6 - \lambda_7\beta_7 \\
& + (\lambda_0\beta_1 + \lambda_1\beta_0 - \lambda_2\beta_4 + \lambda_3\beta_6 + \lambda_4\beta_2 - \lambda_5\beta_7 - \lambda_6\beta_3 - \lambda_7\beta_5)e_1 \\
& + (\lambda_0\beta_2 + \lambda_1\beta_4 + \lambda_2\beta_0 - \lambda_3\beta_5 - \lambda_4\beta_1 + \lambda_5\beta_3 - \lambda_6\beta_7 - \lambda_7\beta_6)e_2 \\
& + (\lambda_0\beta_3 - \lambda_1\beta_6 + \lambda_2\beta_5 + \lambda_3\beta_0 - \lambda_4\beta_7 - \lambda_5\beta_2 + \lambda_6\beta_1 - \lambda_7\beta_4)e_3 \\
& + (\lambda_0\beta_4 + \lambda_1\beta_2 - \lambda_2\beta_1 + \lambda_3\beta_7 + \lambda_4\beta_0 - \lambda_5\beta_6 + \lambda_6\beta_5 + \lambda_7\beta_3)e_{12} \\
& + (\lambda_0\beta_5 + \lambda_1\beta_7 + \lambda_2\beta_3 - \lambda_3\beta_2 + \lambda_4\beta_6 + \lambda_5\beta_0 - \lambda_6\beta_4 + \lambda_7\beta_1)e_{23} \\
& + (\lambda_0\beta_6 - \lambda_1\beta_3 + \lambda_2\beta_7 + \lambda_3\beta_1 - \lambda_4\beta_5 + \lambda_5\beta_4 + \lambda_6\beta_0 + \lambda_7\beta_2)e_{31} \\
& + (\lambda_0\beta_7 + \lambda_1\beta_5 + \lambda_2\beta_6 + \lambda_3\beta_4 + \lambda_4\beta_3 + \lambda_5\beta_1 + \lambda_6\beta_2 + \lambda_7\beta_0)e_{123}
\end{aligned}
\tag{37}
$$

Which produces a new multivector with coefficients for each blade defined as above.

# B   Weighted geometric product in $\mathbb{G}_{(3,0,0)}$

The geometric product between multivectors in $\mathbb{G}_{(3,0,0)}$ consists of 64 interaction pairs that can be grouped into 20 different interaction types, i.e. (scalar/scalar), (scalar/vector), (vector/bivector), etc. For example for the grade-0 (scalar) component of the geometric product between multivectors $A$ and $B$, we can write:

$$
\begin{aligned}
\langle AB \rangle_0 = {} & \lambda_0\beta_0 && (\text{scalar} \cdot \text{scalar}) \\
& + \lambda_1\beta_1 + \lambda_2\beta_2 + \lambda_3\beta_3 && (\text{vector} \cdot \text{vector}) \\
& - \lambda_4\beta_4 - \lambda_5\beta_5 - \lambda_6\beta_6 && (\text{bivector} \cdot \text{bivector}) \\
& - \lambda_7\beta_7 && (\text{trivector} \cdot \text{trivector})
\end{aligned}
\tag{38}
$$

Similarly, for the grade-1 (vector), grade-2 (bivector), and grade-3 (trivector) components:

$$
\begin{aligned}
\langle AB \rangle_1 = {} & \lambda_0\beta_1 e_1 + \lambda_0\beta_2 e_2 + \lambda_0\beta_3 e_3 && (\text{scalar} \cdot \text{vector}) \\
& + \lambda_1\beta_0 e_1 + \lambda_2\beta_0 e_2 + \lambda_3\beta_0 e_3 && (\text{vector} \cdot \text{scalar}) \\
& + \lambda_1\beta_4 e_2 - \lambda_1\beta_6 e_3 - \lambda_2\beta_4 e_1 + \lambda_2\beta_5 e_3 + \lambda_3\beta_6 e_1 - \lambda_3\beta_5 e_2 && (\text{vector} \cdot \text{bivector}) \\
& + \lambda_4\beta_2 e_1 - \lambda_4\beta_1 e_2 + \lambda_5\beta_3 e_2 - \lambda_5\beta_2 e_3 - \lambda_6\beta_3 e_1 + \lambda_6\beta_1 e_3 && (\text{bivector} \cdot \text{vector}) \\
& - \lambda_4\beta_7 e_3 - \lambda_5\beta_7 e_1 - \lambda_6\beta_7 e_2 && (\text{bivector} \cdot \text{trivector}) \\
& - \lambda_7\beta_5 e_1 - \lambda_7\beta_6 e_2 - \lambda_7\beta_4 e_3 && (\text{trivector} \cdot \text{bivector})
\end{aligned}
\tag{39}
$$

$$
\begin{aligned}
\langle AB \rangle_2 = {} & \lambda_0\beta_4 e_{12} + \lambda_0\beta_5 e_{23} + \lambda_0\beta_6 e_{31} && (\text{scalar} \cdot \text{bivector}) \\
& + \lambda_4\beta_0 e_{12} + \lambda_5\beta_0 e_{23} + \lambda_6\beta_0 e_{31} && (\text{bivector} \cdot \text{scalar}) \\
& + \lambda_1\beta_2 e_{12} - \lambda_1\beta_3 e_{31} - \lambda_2\beta_1 e_{12} + \lambda_2\beta_3 e_{23} - \lambda_3\beta_2 e_{23} + \lambda_3\beta_1 e_{31} && (\text{vector} \cdot \text{vector}) \\
& + \lambda_4\beta_6 e_{23} - \lambda_4\beta_5 e_{31} - \lambda_5\beta_6 e_{12} + \lambda_5\beta_4 e_{31} + \lambda_6\beta_5 e_{12} - \lambda_6\beta_4 e_{23} && (\text{bivector} \cdot \text{bivector}) \\
& + \lambda_1\beta_7 e_{23} + \lambda_2\beta_7 e_{31} + \lambda_3\beta_7 e_{12} && (\text{vector} \cdot \text{trivector}) \\
& + \lambda_7\beta_1 e_{23} + \lambda_7\beta_2 e_{31} + \lambda_7\beta_3 e_{12} && (\text{trivector} \cdot \text{vector})
\end{aligned}
\tag{40}
$$

$$
\begin{aligned}
\langle AB \rangle_3 = {} & \lambda_0\beta_7 e_{123} && (\text{scalar} \cdot \text{trivector}) \\
& + \lambda_7\beta_0 e_{123} && (\text{trivector} \cdot \text{scalar}) \\
& + (\lambda_1\beta_5 + \lambda_2\beta_6 + \lambda_3\beta_4)e_{123} && (\text{vector} \cdot \text{bivector}) \\
& + (\lambda_4\beta_3 + \lambda_5\beta_1 + \lambda_6\beta_2)e_{123} && (\text{bivector} \cdot \text{vector})
\end{aligned}
\tag{41}
$$

If we apply separate weights to each of these 20 interaction terms and collect terms with the same

weights, we can derive a weighted geometric product that is equivariant to $O(3)$:

$$
\begin{aligned}
\mathrm{GP}(A, B)_w =& \\
& w_s\lambda_0\beta_0 + w_v(\lambda_1\beta_1 + \lambda_2\beta_2 + \lambda_3\beta_3) - w_b(\lambda_4\beta_4 + \lambda_5\beta_5 + \lambda_6\beta_6) - w_t\lambda_7\beta_7 \\
&+ \big(w_{sv}\lambda_0\beta_1 + w_{vs}\lambda_1\beta_0 + w_{vb}(-\lambda_2\beta_4 + \lambda_3\beta_6) + w_{bv}(\lambda_4\beta_2 - \lambda_6\beta_3) - w_{bt}\lambda_5\beta_7 - w_{tb}\lambda_7\beta_5\big)e_1 \\
&+ \big(w_{sv}\lambda_0\beta_2 + w_{vs}\lambda_2\beta_0 + w_{vb}(\lambda_1\beta_4 - \lambda_3\beta_5) + w_{bv}(-\lambda_4\beta_1 + \lambda_5\beta_3) - w_{bt}\lambda_6\beta_7 - w_{tb}\lambda_7\beta_6\big)e_2 \\
&+ \big(w_{sv}\lambda_0\beta_3 + w_{vs}\lambda_3\beta_0 + w_{vb}(-\lambda_1\beta_6 + \lambda_2\beta_5) + w_{bv}(-\lambda_5\beta_2 - \lambda_6\beta_1) - w_{bt}\lambda_4\beta_7 - w_{tb}\lambda_7\beta_4\big)e_3 \qquad (42) \\
&+ \big(w_{sb}\lambda_0\beta_4 + w_{bs}\lambda_4\beta_0 + w_{vv}(\lambda_1\beta_2 - \lambda_2\beta_1) + w_{bb}(-\lambda_5\beta_6 + \lambda_6\beta_5) + w_{vt}\lambda_3\beta_7 + w_{tv}\lambda_7\beta_3\big)e_{12} \\
&+ \big(w_{sb}\lambda_0\beta_5 + w_{bs}\lambda_5\beta_0 + w_{vv}(\lambda_2\beta_3 - \lambda_3\beta_2) + w_{bb}(\lambda_4\beta_6 - \lambda_6\beta_4) + w_{vt}\lambda_1\beta_7 + w_{tv}\lambda_7\beta_1\big)e_{23} \\
&+ \big(w_{sb}\lambda_0\beta_6 + w_{bs}\lambda_6\beta_0 + w_{vv}(-\lambda_1\beta_3 + \lambda_3\beta_1) + w_{bb}(-\lambda_4\beta_5 + \lambda_5\beta_4) + w_{vt}\lambda_2\beta_7 + w_{tv}\lambda_7\beta_2\big)e_{31} \\
&+ \big(w_{st}\lambda_0\beta_7 + w_{tvb}(\lambda_1\beta_5 + \lambda_2\beta_6 + \lambda_3\beta_4) + w_{tbv}(\lambda_4\beta_3 + \lambda_5\beta_1 + \lambda_6\beta_2) + w_{ts}\lambda_7\beta_0\big)e_{123}
\end{aligned}
$$

The weights can be initialized in many different ways. In our experiments, we chose to initialize them using a normal distribution with zero mean and standard deviation $1/\sqrt{8}$. Additionally, one could add a bias term to the scalar component. Adding bias to the trivector component maintains rotation equivariance, but breaks equivariance with respect to reflection.

## C   Including the geometric product in the message block

We test two variations of an alternative message block that utilizes the weighted geometric product. Here, we describe the variation between sender and receiver pairs. For each edge $(i, j)$ we compute the weighted geometric product:

$$A_{ij} = \mathrm{GP}(A_i, A_j)_w \qquad (43)$$

where $A_i$ and $A_j$ are the sender and receiver multivector states. The resulting multivector $A_{ij}$ is then passed through a linear layer:

$$A'_{ij} = \mathbf{W} \cdot A_{ij} \qquad (44)$$

The grades of this transformed multivector are then gated and aggregated analogously to the base architecture formulation instead of using the grades of the sender multivector. One exception is that the scalar message also uses the multivector grade:

$$\mathbf{m}_i^s = \sum_{j \in \mathcal{N}(i)} \mathbf{g}_{ij}^{(s)} \circ \langle A'_{ij} \rangle_0 \qquad (45)$$

$$\mathbf{m}_i^v = \sum_{j \in \mathcal{N}(i)} \mathbf{g}_{ij}^{(v)} \circ \langle A'_{ij} \rangle_1 + \mathbf{g}_{ij}^{(d)} \circ \frac{\mathbf{r}_{ij}}{\|\mathbf{r}_{ij}\|} \qquad (46)$$

$$\mathbf{m}_i^b = \sum_{j \in \mathcal{N}(i)} \mathbf{g}_{ij}^{(b)} \circ \langle A'_{ij} \rangle_2 \qquad (47)$$

$$\mathbf{m}_i^t = \sum_{j \in \mathcal{N}(i)} \mathbf{g}_{ij}^{(t)} \circ \langle A'_{ij} \rangle_3 \qquad (48)$$

The variation that takes the weighted geometric product between the sender state and a linear projection of itself uses the same formulation as above, but with the weighted geometric product input changed accordingly.

## D   Hyperparameters for training

The same hyperparameters are used across all experiments, with the exception that ablation and architecture variation studies use a channel dimension of $F = 64$, whereas the main evaluation uses $F = 128$ for all targets except for $\langle R^2 \rangle$, which still uses $F = 64$. Additionally, for the $\epsilon_{\mathrm{LUMO}}$ and $\Delta\epsilon$ targets, we lowered the gradient-clipping max-norm from 1.0 to 0.5 on seed 2 to obtain stable training as the first runs on this seed resulted in unstable training. Furthermore, For the $U$ target, seed 2 could not be trained stably when using an output MLP, regardless of the gradient-clipping threshold. All experiments are trained with $T = 4$ message passing rounds.

Two types of random seeds are used in the experiments. A global random seed is fixed to zero in all runs to ensure reproducibility of stochastic elements such as model weight initialization and any other

**Table D.1.** Hyperparameters used for training.

| Hyperparameter | Value |
|---|---|
| `batch_size` | 100 |
| `learning rate` (initial) | 5e-4 |
| `minimum learning rate` | 1e-6 |
| `weight decay` | 0.01 |
| `patience` (lr decay) | 5 |
| `patience` (early stopping) | 30 |
| `alpha` (EMA smoothing) | 0.9 |
| `train/val/test split sizes` | [110000, 10000, 10831] |
| `Global seed` | 0 |
| `Data split seed (ablation/variation)` | 0 |
| `Data split seeds (main evaluation)` | [1, 2, 3] |

random operations during training. In addition, a data split seed controls the shuffling used to generate the train/validation/test splits (with fixed split sizes but different assignments). For ablation and variation experiments, only split seed 0 is used. For the main evaluation, results are averaged over split seeds 1, 2, and 3, corresponding to three distinct data splits. The set of hyperparameters is listed in Table D.1.

# E    Descriptions of each architecture addition/ablation.

Table E.1 describes each addition/ablation for the architecture variation and ablation study in detail.

# F    Results per individual seed/split

Table F.1 shows the MAE of GA-GNN on each random data split across the twelve QM9 targets as well as the mean and standard deviation.

# G    Computational Complexity Analysis

Table G.1 reports the number of trainable parameters as well as the training time of GA-GNN in terms of average number of epochs, time per epoch and total GPU days. We show results for the four variants of GA-GNN used in the main evaluation for the targets used in the architecture selection study. For reference, we compare these results to the computational cost analysis reported in [20].

**Table E.1.** Overview and description of each architectural addition and ablation.

| Additions | Description |
|---|---|
| Sender/receiver GP | Using the geometric product between sender and receiver nodes in the message block as described in Appendix C. |
| Sender/copy GP | Using the geometric product between sender nodes and a linear projection of themselves in the message block as described in Appendix C. |
| 3 GP in update block | Setting $N = 3$ in Eq. 20 and Eq. 22 to extend the update block with three successive geometric product layers instead of two. And we set $X_{2,i} = Y_{1,i}$ for the third geometric product. |
| Grade-wise linear layers | Replace all shared linear layers with grade-wise linear layers, as described in Eq. 10. |
| **Ablations** | |
| Removal of second GP | Remove the second geometric product layer from the update block, i.e. set $N = 1$ in Eq. 20 and Eq. 22. |
| Non-weighted GPs | Remove learnable scalar weights from the geometric product operation, making the product unweighted and implemented as in Appendix A. |
| No output networks | Removal of the gated equivariant blocks for dipole moment ($\mu$) prediction, and two-layer MLP's for scalar and ($\langle R^2 \rangle$) prediction. As described in section 3.1. |
| Trivectors initialized as 0 | Initialize the trivector component of the node state to zero instead of using learned embeddings of the atom type. |
| Shared update MLP | Use a single MLP shared across all atom types, rather than atom-type specific MLP's in the update block to compute residual update gates (i.e. removing $z_i$ index in Eq. 21. |
| **Base architecture** | The default architecture as described in Section 3.1. |

**Table F.1.** MAE of GA-GNN for each QM9 target on each random split with mean and standard deviation.

| Target | Split 1 | Split 2 | Split 3 | Mean | Std |
|---|---|---|---|---|---|
| $\epsilon_{\text{HOMO}}$ | 20.3650 | 21.3509 | 20.4838 | 20.7332 | 0.5382 |
| $\epsilon_{\text{LUMO}}$ | 17.7386 | 17.7401 | 17.2431 | 17.5739 | 0.2865 |
| $\Delta\epsilon$ | 35.5387 | 35.7047 | 35.7065 | 35.6500 | 0.0964 |
| $\mu$ | 0.0109 | 0.0108 | 0.0109 | 0.0109 | 0.0001 |
| $\langle R^2 \rangle$ (grade-wise linear layers) | 0.0585 | 0.0676 | 0.0633 | 0.0631 | 0.0046 |
| $\langle R^2 \rangle$ (shared linear layers) | 0.0633 | 0.0622 | 0.0677 | 0.0644 | 0.0029 |
| $\alpha$ | 0.0461 | 0.0462 | 0.0439 | 0.0454 | 0.0013 |
| ZPVE | 1.1860 | 1.1669 | 1.1764 | 1.1764 | 0.0096 |
| $U_0$ | 6.1722 | 6.1689 | 6.2838 | 6.2083 | 0.0654 |
| $U$ | 6.3403 | 5.9967 | 6.1578 | 6.1649 | 0.1719 |
| $H$ | 6.0935 | 6.0775 | 6.1352 | 6.1021 | 0.0298 |
| $G$ | 7.3221 | 7.3762 | 6.9298 | 7.2094 | 0.2436 |
| $c_v$ | 0.0231 | 0.0232 | 0.0228 | 0.0230 | 0.0002 |

**Table G.1.** Training performance and model sizes for GA-GNN architectures compared to Equiformer, Equiformer V2 and GotenNet variants.

| Model | Batch size | Avg. total epochs | Avg. time per epoch (s) | Avg. GPU days | Trainable parameters |
|---|---|---|---|---|---|
| Equiformer | 128 | - | 425 | 1.48 | 3.5M |
| EquiformerV2 | 64 | - | 821 | 2.85 | 11.2M |
| EquiformerV2 | 48 | - | 847 | 2.94 | 11.2M |
| GotenNet-B | 32 | - | 180 | 1.15 | 9.2M |
| GotenNet-S | 32 | - | 117 | 0.75 | 6.1M |
| GA-GNN_alpha | 100 | 508 | 169 | 0.99 | 10.1M |
| GA-GNN_mu | 100 | 498 | 185 | 1.06 | 10.2M |
| GA-GNN_R2 | 100 | 1000 | 127 | 1.46 | 2.8M |
| GA-GNN_homo | 100 | 321 | 171 | 0.63 | 10.1M |

