# OpenReview forum: "Design and Evaluation of a Geometric Algebra-Based Graph Neural Network for Molecular Property Prediction"
_NLDL.org/2026/Conference — NLDL 2026 Spotlight_

### Official Review · Reviewer_ydoS · 2025-09-18

**Rating:** 4
**Confidence:** 3

**Summary:**

The paper proposes a new type of GNN that exploits "geometric algebra", which allows maintaining higher-order geometric structure information during the message passing.

**Strengths:**

The paper is well written: the complexity of both the new theoretical framework and the derived GNN is presented very well and in clear manner. The experiments are also comprehensive, and contain both extensive ablations and design analysis, and a large benchmark comparison. The results are transparent and clear.

The idea of GA in GNNs is innovative and has future potential.

The method's performance is competetive.

**Weaknesses:**

While the GA formalism intuitively provides new viewpoint to 3D structures, it wasn't totally clear what is actually the benefit of this formalism. What information does the GA handle that vanilla GNN's can't do? In what way do we expect the GA formalism to be better? I'm not totally sure why eg. transferring information between the vector and trivector channels is useful. The paper would become stronger by giving a clearer intuition.

The results are not state of the art, with GotenNet being way better. The analysis of why the GA model does not work as well is quite superficial, and could have been expanded.

**Justification:**

This is a very nicely written paper about a very nicely executed research project, with a novel idea and good results. The pros clearly outweigh the main negative of not achieving SOTA.

---

> ### Author Rebuttal · Authors · 2025-10-22
>
> We thank the reviewer for the thoughtful and constructive feedback. We address comments regarding the intuition behind the GA formalism and the analysis of the results below.
>
> > W1. While the GA formalism intuitively provides a new viewpoint to 3D structures, it wasn't totally clear what is actually the benefit of this formalism. What information does the GA handle that vanilla GNN's can't do? In what way do we expect the GA formalism to be better? I'm not totally sure why eg. transferring information between the vector and trivector channels is useful. The paper would become stronger by giving a clearer intuition.
>
> The multivector product is a natural equivariant operation, and parameterizing it allows interactions between scalars, vectors, bivectors, and trivectors in a unified framework. This enables the network to capture geometric relationships—such as orientations, areas, and volumes—that vanilla GNNs cannot explicitly represent. In particular, transferring information between vector and trivector channels allows volumetric and orientational information to propagate through the network. We will clarify this intuition in the revised manuscript.
>
> > W2. The results are not state of the art, with GotenNet being way better. The analysis of why the GA model does not work as well is quite superficial, and could have been expanded.
>
> While our GA-based model ranks above what was considered state of the art up until mid-2025, it is indeed interesting to examine in more detail the factors behind the apparent large performance improvement of GotenNet. We will provide a more detailed discussion of this comparison in the revised manuscript.

---

### Official Review · Reviewer_n5a5 · 2025-10-03
**Review of GA-GNN**

**Rating:** 2
**Confidence:** 5

**Summary:**

This paper introduces GA-GNN, a graph neural network that leverages Geometric Algebra (GA) multivector representations for molecular property prediction. The work represents a solid technical contribution demonstrating that GA-based architectures can achieve competitive performance on molecular benchmarks, though it falls short of establishing clear advantages over existing methods.

**Strengths:**

- The application of Geometric Algebra to molecular property prediction via GNNs is relatively unexplored. The paper successfully adapts CGENN layers to the molecular domain, providing a concrete instantiation of how multivector representations can be integrated into message-passing frameworks.
- The systematic evaluation of architectural variations (Table 2) provides valuable insights into design choices. The finding that removing output networks improves performance on 3/4 targets is particularly interesting and suggests the multivector representation may naturally encode sufficient information for certain properties.
- Competitive empirical results: GA-GNN achieves strong performance on dipole moment (0.011 D) and electronic spatial extent (0.063 α₀²), ranking 3rd among compared methods. The model's particular strength on dipole moment aligns well with the vector nature of this property.
- The background on Geometric Algebra is well-written and accessible. The architectural diagrams (Figures 1-2) effectively communicate the model design.

**Weaknesses:**

- Testing on only 4/19 QM9 targets significantly weakens the empirical contribution. The selective evaluation makes it difficult to assess the general applicability of the approach. A complete evaluation would strengthen claims about the method's effectiveness.
-  While the authors acknowledge "relatively high computational cost," no concrete runtime comparisons or complexity analysis is provided. Given that geometric products scale with O(d²) where d is the multivector dimension, this is a significant practical limitation that deserves quantitative analysis.
-  The paper doesn't establish when or why GA representations should be preferred over existing tensor-based approaches. GA-GNN ranks 5th on average among compared models, with GotenNet B achieving superior performance across all evaluated targets. The theoretical benefits of unified multivector representations aren't clearly translated to practical advantages.
- The architecture selection study (Section 4.1) uses single training runs, making it difficult to distinguish meaningful patterns from noise. The authors acknowledge this limitation but still use these results to guide architectural choices. Additionally, the main evaluation uses only 3 random splits - more would provide better statistical confidence.
- Recent models like Equiformer V2 and GotenNet significantly outperform GA-GNN. The paper would benefit from deeper analysis of why these models succeed where GA-GNN struggles, and what specific advantages GA might offer in other contexts.
- The focus on G(3,0,0) is reasonable but the paper doesn't explore whether different GA spaces might be better suited for molecular tasks. The brief mention of G(3,0,1) in the discussion feels like an afterthought rather than a principled investigation.
- The paper could benefit from discussing why certain properties (dipole, R²) align better with GA representations.

Some other parts:
- The initialization of trivectors with learnable embeddings seems unmotivated given that molecules don't naturally have trivector properties. The ablation showing improved performance with zero initialization for ϵ_HOMO supports this concern.
- The weighted geometric product introduces 20 learnable parameters per layer, but the ablation shows removing these weights only modestly degrades performance, suggesting the benefit may not justify the complexity.
- The paper claims architectural flexibility but then shows different optimal architectures for different targets, somewhat undermining this claimed advantage.

**Justification:**

This is a technically sound paper that demonstrates the feasibility of using Geometric Algebra for molecular property prediction. The work makes a contribution to understanding how alternative mathematical frameworks can be applied to molecular modeling. However, the limited evaluation scope, lack of clear practical advantages, and absence of computational cost analysis prevent this from being a strong contribution. The paper would be significantly strengthened by complete QM9 evaluation, runtime comparisons, and majorly when GA-based approaches should be preferred over existing methods.

---

> ### Author Rebuttal · Authors · 2025-10-22
>
> We thank the reviewer for the thorough and thoughtful assessment of our work and for the constructive feedback. We address the identified weaknesses in detail below.
>
> > W1. Testing on only 4/19 QM9 targets significantly weakens the empirical contribution. The selective evaluation makes it difficult to assess the general applicability of the approach. A complete evaluation would strengthen claims about the method's effectiveness.
>
> We agree that expanding the evaluation to include additional targets and datasets would further strengthen the study. If the paper is accepted, we will extend our experiments in the revised manuscript to include the twelve QM9 targets, aligning with the evaluation setup used by other models such as GotenNet. This will provide a more comprehensive comparison and help validate GA-GNN’s robustness across a broader range of molecular properties. We have already performed preliminary experiments on each additional target, using a fixed set of hyperparameters. In these initial tests, which is based on one run per target, we employed the same architecture across all additional targets consisting of one geometric product layer, no output network, and trivectors initialized to zero rather than learned embeddings. Despite the simplicity of the setup, the results are competitive with Equiformer v2 and MACE.
>
> | Model   | ε_HOMO (meV) | ε_LUMO (meV) | Δε (meV) | μ (D)  | R² (α₀²₀) | α (α₀³₀) | ZPVE (meV) | U₀ (meV) | U (meV) | H (meV) | G (meV) | cᵥ (cal/mol·K) |
> |----------|---------------|---------------|-----------|--------|-------------|------------|-------------|-----------|-----------|-----------|-----------|----------------|
> | GA-GNN   | 21            | 21            | 39        | 0.011  | 0.063       | 0.045      | 1.16        | 6.3       | 6.1       | 6.2       | 7.5       | 0.024          |
>
> > W2. While the authors acknowledge "relatively high computational cost," no concrete runtime comparisons or complexity analysis is provided. Given that geometric products scale with O(d²) where d is the multivector dimension, this is a significant practical limitation that deserves quantitative analysis.
>
> Compared with state-of-the-art models, our model is not computationally heavy. In fact, it is comparable to GotenNet-B and has fewer parameters and a faster runtime than Equiformer v2 as shown in the table below:
>
> | Model           | Batch size | Avg. total epochs | Avg. time per epoch (s) | Avg. GPU days | Trainable parameters |
> |-----------------|------------|--------------------|---------------------------|---------------|------------------------|
> | Equiformer      | 128        | -                  | 425                       | 1.48          | 3.5M                   |
> | EquiformerV2    | 64         | -                  | 821                       | 2.85          | 11.2M                  |
> | EquiformerV2    | 48         | -                  | 847                       | 2.94          | 11.2M                  |
> | GotenNet-B      | 32         | -                  | 180                       | 1.15          | 9.2M                   |
> | GotenNet-S      | 32         | -                  | 117                       | 0.75          | 6.1M                   |
> | GA-GNN_alpha    | 100        | 508                | 169                       | 0.99          | 10.1M                  |
> | GA-GNN_mu       | 100        | 498                | 185                       | 1.06          | 10.2M                  |
> | GA-GNN_R2       | 100        | 1000               | 127                       | 1.46          | 2.8M                   |
> | GA-GNN_homo     | 100        | 321                | 171                       | 0.63          | 10.1M                  |
>
> These results are based on the reported complexity cost analysis in [1] and the runs for the four variants of GA-GNN used in our main evaluation. We will clarify these points in the discussion and include detailed runtime comparisons in the appendix in the final version.
>
> [1] Aykent, S. and Xia, T. (2025). *GotenNet: Rethinking Efficient 3D Equivariant Graph Neural Networks.*
> The Thirteenth International Conference on Learning Representations (ICLR).
>
> > W3. The paper doesn't establish when or why GA representations should be preferred over existing tensor-based approaches. GA-GNN ranks 5th on average among compared models, with GotenNet B achieving superior performance across all evaluated targets. The theoretical benefits of unified multivector representations aren't clearly translated to practical advantages.
>
> We agree, and we also find it an interesting open question whether—and under what conditions—GA representations offer advantages over, for example, irreducible representations of SO(3) or O(3). While in certain special cases GA and irrep-based representations can be nearly equivalent, their geometric interpretations differ, and the natural extensions of GA to other algebras, such as CGA and PGA, open promising new directions.
>
> We would also like to note that, while ranking fifth on average may not seem outstanding, our model achieves the second-best average ranking overall, surpassed only by GotenNet-B, and thus outperforms what was considered state of the art up until mid-2025.
>
> > W4. The architecture selection study (Section 4.1) uses single training runs, making it difficult to distinguish meaningful patterns from noise. The authors acknowledge this limitation but still use these results to guide architectural choices. Additionally, the main evaluation uses only 3 random splits - more would provide better statistical confidence.
>
> We agree that using single training runs in the architecture selection study is suboptimal. Our goal was to explore a broad range of design choices and identify general trends rather than make statistically definitive claims about optimal architectures. This work focuses on demonstrating the feasibility and potential of a GA-based GNN architecture, not exhaustive optimization for peak QM9 performance.
>
> Regarding the use of three random splits in the main evaluation, we agree that more splits would improve statistical confidence. Nonetheless, three independent splits already exceed the standard practice for QM9 in many highly regarded papers. Moreover, the small standard deviations across runs indicate that our results are stable.
>
> > W5. Recent models like Equiformer V2 and GotenNet significantly outperform GA-GNN. The paper would benefit from deeper analysis of why these models succeed where GA-GNN struggles, and what specific advantages GA might offer in other contexts.
>
> While GotenNet outperforms GA-GNN, it is inaccurate to claim the same for Equiformer V2. GA-GNN performs better on two of four targets and has a lower average ranking (5.0) than Equiformer V2 (5.75), ranking second overall behind GotenNet (1.50). We agree that a deeper analysis of GA-GNN’s strengths and limitations compared to transformer-based models like GotenNet and Equiformer V2 would strengthen the paper, and we will include this in the revision.
>
> > W6. The focus on G(3,0,0) is reasonable but the paper doesn't explore whether different GA spaces might be better suited for molecular tasks. The brief mention of G(3,0,1) in the discussion feels like an afterthought rather than a principled investigation.
>
> We agree that exploring different GA spaces is an interesting direction for further research. While our brief mention of G(3,0,1) in the discussion was intended to highlight this direction, we recognize that a systematic comparison would strengthen the study.
>
> > W7. The paper could benefit from discussing why certain properties (dipole, R²) align better with GA representations.
>
> We agree. We expect that GA representations are particularly relevant for properties with a clear geometric interpretation, such as the norm of the dipole moment or the electronic spatial extent. We will expand the discussion to highlight the geometric interpretation of these properties.
>
> **Some other parts:**
> > 1: The initialization of trivectors with learnable embeddings seems unmotivated given that molecules don't naturally have trivector properties. The ablation showing improved performance with zero initialization for ϵ_HOMO supports this concern.
>
> We agree with this concern. We included trivector initialization in the ablation study to evaluate its effect empirically. The results indeed suggest that it is not beneficial except for the case of R², which is noticeably worse with trivectors initialized as zero.
>
> > 2: The weighted geometric product introduces 20 learnable parameters per layer, but the ablation shows removing these weights only modestly degrades performance, suggesting the benefit may not justify the complexity.
>
> We agree with this point, and it is reflected in the paper: we note on lines 449-454 that the ablation study suggests most of the benefit comes from the structure of the geometric product itself, with the weights mainly refining the computation. That said, the additional weights have negligible overhead compared to the geometric product operation, and since they improve performance, we included them in the architectures for the main evaluation.
>
> > 3: The paper claims architectural flexibility but then shows different optimal architectures for different targets, somewhat undermining this claimed advantage.
>
> While the model is not flexible in the sense that a single architecture is optimal across all targets, the claim of flexibility is related to the idea that multivector embeddings and geometric product layers can handle targets of different geometric nature without the need for task-specific architectural components (e.g. specialized blocks for vector targets as in PaiNN). We will make this distinction more clear in the revised manuscript.

---

### Official Review · Reviewer_Ngnf · 2025-10-07

**Rating:** 2
**Confidence:** 3

**Summary:**

The paper proposes GA-GNN. The model leverages Geometric Algebra (GA) to represent node features as unified "multivectors," which combine scalars, vectors, and higher-dimensional geometric elements. GA-GNN uses sequences of geometric product layers as its core update mechanism and achieves competitive performance on the QM9 benchmark

**Strengths:**

1. The use of Geometric Algebra (GA) provides a unified and mathematically robust framework for representing geometric entities.
2. The model's core operation, the geometric product, is naturally equivariant to rotation and reflection (O(3)) transformations. This ensures that learned features transform consistently with the molecule's geometry.

**Weaknesses:**

1. The authors note that the approach has a high computational cost and parameter count, particularly due to the weighted geometric products and atom-type-specific MLPs in the update block. Experiments showed that including geometric products in the message block significantly increased cost without a justifiable performance gain.
2. The model was evaluated on only four of the 19 available regression targets in the QM9 dataset. A broader evaluation on all QM9 targets and other domains would be needed to fully assess the model's capabilities.
3. Some missing related work on molecular property prediction. [1-4]

[1]. Yu, Zhaoning, and Hongyang Gao. "Molecular representation learning via heterogeneous motif graph neural networks." International conference on machine learning. PMLR, 2022.

[2]. Chen, Dingshuo, et al. "Uncovering neural scaling laws in molecular representation learning." Advances in Neural Information Processing Systems 36 (2023): 1452-1475.

[3]. Kim, Suyeon, et al. "Learning topology-specific experts for molecular property prediction." Proceedings of the AAAI Conference on Artificial Intelligence. Vol. 37. No. 7. 2023.

[4]. Guo, Minghao, et al. "Hierarchical grammar-induced geometry for data-efficient molecular property prediction." International Conference on Machine Learning. PMLR, 2023.

**Justification:**

This paper offers an interesting application of geometric algebra to molecular property prediction. Yet, given the high computational cost and the narrow evaluation, the submission is not ready for acceptance; a strengthened empirical study and efficiency analysis would be needed.

---

> ### Author Rebuttal · Authors · 2025-10-22
>
> We thank the reviewer for the assessment of our paper and the constructive feedback. We address the concerns regarding computational cost, evaluation scope and missing related work below.
>
> > W1. The authors note that the approach has a high computational cost and parameter count, particularly due to the weighted geometric products and atom-type-specific MLPs in the update block. Experiments showed that including geometric products in the message block significantly increased cost without a justifiable performance gain.
>
> We would like to clarify that the geometric products in the message blocks (per edge) incur a substantially higher computational cost without yielding a justifiable performance gain, whereas using geometric products only in the update blocks (per node) results in strong performance. This is one of the findings of our paper. While atom-type-specific MLPs do increase the number of parameters, this is a common design choice, and our ablation studies confirm that the same holds true in our setting.
>
> Moreover, compared with state-of-the-art models, our model is not computationally heavy. In fact, it is comparable to GotenNet-B and has fewer parameters and a faster runtime than Equiformer v2 as shown in the table below:
>
> | Model           | Batch size | Avg. total epochs | Avg. time per epoch (s) | Avg. GPU days | Trainable parameters |
> |-----------------|------------|--------------------|---------------------------|---------------|------------------------|
> | Equiformer      | 128        | -                  | 425                       | 1.48          | 3.5M                   |
> | EquiformerV2    | 64         | -                  | 821                       | 2.85          | 11.2M                  |
> | EquiformerV2    | 48         | -                  | 847                       | 2.94          | 11.2M                  |
> | GotenNet-B      | 32         | -                  | 180                       | 1.15          | 9.2M                   |
> | GotenNet-S      | 32         | -                  | 117                       | 0.75          | 6.1M                   |
> | GA-GNN_alpha    | 100        | 508                | 169                       | 0.99          | 10.1M                  |
> | GA-GNN_mu       | 100        | 498                | 185                       | 1.06          | 10.2M                  |
> | GA-GNN_R2       | 100        | 1000               | 127                       | 1.46          | 2.8M                   |
> | GA-GNN_homo     | 100        | 321                | 171                       | 0.63          | 10.1M                  |
>
> These results are based on the reported complexity cost analysis in [1] and the runs for the four variants of GA-GNN used in our main evaluation. We will clarify these points in the discussion and include detailed runtime comparisons in the appendix in the final version.
>
> > W2. The model was evaluated on only four of the 19 available regression targets in the QM9 dataset. A broader evaluation on all QM9 targets and other domains would be needed to fully assess the model's capabilities.
>
> We agree that expanding the evaluation to include additional targets and datasets would further strengthen the study. If the paper is accepted, we will extend our experiments in the revised manuscript to include the twelve QM9 targets, aligning with the evaluation setup used by other models such as GotenNet. This will provide a more comprehensive comparison and help validate GA-GNN’s robustness across a broader range of molecular properties. We have already performed preliminary experiments on each additional target, using a fixed set of hyperparameters. In these initial tests, which is based on one run per target, we employed the same architecture across all additional targets consisting of one geometric product layer, no output network, and trivectors initialized to zero rather than learned embeddings. Despite the simplicity of the setup, the results are competitive with Equiformer v2 and MACE.
>
> | Model   | ε_HOMO (meV) | ε_LUMO (meV) | Δε (meV) | μ (D)  | R² (α₀²₀) | α (α₀³₀) | ZPVE (meV) | U₀ (meV) | U (meV) | H (meV) | G (meV) | cᵥ (cal/mol·K) |
> |----------|---------------|---------------|-----------|--------|-------------|------------|-------------|-----------|-----------|-----------|-----------|----------------|
> | GA-GNN   | 21            | 21            | 39        | 0.011  | 0.063       | 0.045      | 1.16        | 6.3       | 6.1       | 6.2       | 7.5       | 0.024          |
>
> > W3. Some missing related work on molecular property prediction. [1-4]
>
> We thank the reviewer for pointing out these additional works on molecular property prediction. We will carefully review and incorporate them into the revised manuscript to ensure a more complete coverage of related research.
>
> [1] Aykent, S. and Xia, T. (2025). *GotenNet: Rethinking Efficient 3D Equivariant Graph Neural Networks.*
> The Thirteenth International Conference on Learning Representations (ICLR).

---

### Official Review · Reviewer_28Uk · 2025-10-08
**A Novel Geometric Algebra GNN: Promising architecture and strong results, but limited scope and nconsistent unification**

**Rating:** 4
**Confidence:** 4
**Final Rating:** 4
**Final Confidence:** 4

**Summary:**

The paper addresses the challenge of designing O(3)-equivariant graph neural networks to predict the properties of molecules, which are represented as graphs embedded in 3D space. The goal is to create architectures that can handle geometric transformations like rotation and reflection accurately.

The paper introduces GA-GNN, a novel architecture that represents node (atom) states as unified multivectors from Geometric Algebra. This is a departure from prior work like PaiNN, which uses separate scalar and vector feature channels. The model's core update mechanism is a sequence of geometric product layers, adapted from Clifford Group Equivariant Neural Networks, which mix information across different geometric grades (scalar, vector, bivector, trivector).

The results are obtained on QM9 dataset with four specific regression targets. This shows competitive performance with recent SOTA. They also show simplification of architectural design, for example, by enabling flexible and effective readout layers that do not require specialized, target-specific networks.

**Strengths:**

1. Novel and Sound Architecture

The architecture uses Geometric Algebra for molecular property prediction; the transition to a unified multivector is an elegant approach to encoding equivariant features. Being the first application of a GA-based model to molecular property prediction on QM9, this work successfully demonstrates the feasibility of this approach.

2. Thorough Ablation study

  In addition to proper results, the ablations presented in 4.1 and Table 2 show a rigorous approach. By systematically investigating a wide range of design choices, especially the removal of specialized output networks, the paper substantiates the choices in the architecture's framework, as well as supports the claim of simplification.

3. Strong performance:

The proposed model achieves competitive performance, if not superior to, many highly-regarded models such as PaiNN and MACE.

4. Clear presentation

It is well written and organized. The topics are given a clear introduction, building a strong foundation for a broader machine learning audience. Figures also align with the framework being discussed.

**Weaknesses:**

1. Not establishing SOTA.

Though competitive with recent SOTAs, the model does not establish a new SOTA on any of the four reported targets. It is clearly surpassed by GotenNet B, considering overall performance. This context is crucial for positioning the paper's contribution as an exploration of a promising new methodology rather than a breakthrough in performance.

2. Limited Scope of Evaluation.

Considering that the results are only validated on the QM9 dataset with a total of four selected targets, the remaining properties and other datasets are left unexplored.

3. Target-specific model choices and 'unified framework' conflict:

"unified framework, requiring only minimal adjustments across targets" seems like an indecisive claim, since the need for target-specific tuning weakens the claim of a single, simple, and unified architecture.

4. Concerns Regarding Computational Cost:

As the author(s) acknowledge in the Discussion section, a practical downside of the proposed model is its "relatively high computational cost and parameter count." The main high-performing models presented are computationally intensive, which could be a barrier to adoption for larger-scale applications. Have authors figured out any way to optimize the already proposed lighter variants or any new ones?

**Final Justification:**

As an exploration of a promising new methodology, the paper performs reasonably well, although it is limited by evaluation on a single dataset and a lack of foresight into broader applications. The rebuttal helped clarify several concerns, and the promised revision includes additional targets from the same dataset. While the weaknesses are not entirely resolved, the contribution toward developing a new framework is meaningful and justifies acceptance.

**Justification:**

- introduces a novel and well-motivated application of Geometric Algebra to the important problem of molecular property prediction.
- The proposed GA-GNN architecture is presented clearly, and its effectiveness is supported by a rigorous experimental study featuring thorough ablations and strong, competitive results on the QM9 benchmark.
- The work does not establish a new state-of-the-art and exhibits some limitations regarding the scope of its evaluation and the consistency of its "unified framework" claim.

Considering its clear conceptual contribution, methodical execution, and insightful findings, the paper is a valuable contribution to the literature.

---

> ### Author Rebuttal · Authors · 2025-10-22
>
> We thank the reviewer for the thoughtful and encouraging assessment of our work. We address below the concerns regarding not establishing SOTA, evaluation scope, the claim of a unified framework, and concerns about computational cost.
>
> > W1. Not establishing SOTA:
> Though competitive with recent SOTAs, the model does not establish a new SOTA on any of the four reported targets. It is clearly surpassed by GotenNet B, considering overall performance. This context is crucial for positioning the paper's contribution as an exploration of a promising new methodology rather than a breakthrough in performance.
>
> We agree with the reviewer that GA-GNN does not present a breakthrough in performance, as it does not establish a new SOTA on QM9. The primary goal of this work is to demonstrate the feasibility and potential of a Geometric Algebra-based architecture for molecular property prediction, rather than to optimize for peak benchmark performance. We believe that the results validate this feasibility, with GA-GNN achieving the second-highest average rank across all compared models, only surpassed by GotenNet. This is despite the fact that we do not conduct an extensive hyperparameter search. We will clarify this in the revised version to emphasize that GA-GNN represents a conceptual and architectural contribution rather than a hyperparameter-optimized SOTA benchmark.
>
> > W2. Limited Scope of Evaluation: Considering that the results are only validated on the QM9 dataset with a total of four selected targets, the remaining properties and other datasets are left unexplored.
>
> We agree that expanding the evaluation to include additional targets and datasets would further strengthen the study. For the revised manuscript, we will extend our experiments to the twelve QM9 targets, aligning with the evaluation setup used by other models such as GotenNet. This will provide a more comprehensive comparison and help validate GA-GNN’s robustness across a broader range of molecular properties. We have already performed preliminary experiments on each additional target, using a fixed set of hyperparameters. In these initial tests, which is based on one run per target, we employed the same architecture across all additional targets consisting of one geometric product layer, no output network, and trivectors initialized to zero rather than learned embeddings. Despite the simplicity of the setup, the results are competitive with  Equiformer v2 and MACE.
>
> | Model   | ε_HOMO (meV) | ε_LUMO (meV) | Δε (meV) | μ (D)  | R² (α₀²₀) | α (α₀³₀) | ZPVE (meV) | U₀ (meV) | U (meV) | H (meV) | G (meV) | cᵥ (cal/mol·K) |
> |----------|---------------|---------------|-----------|--------|-------------|------------|-------------|-----------|-----------|-----------|-----------|----------------|
> | GA-GNN   | 21            | 21            | 39        | 0.011  | 0.063       | 0.045      | 1.16        | 6.3       | 6.1       | 6.2       | 7.5       | 0.024          |
>
> > W3. Target-specific model choices and 'unified framework' conflict: "unified framework, requiring only minimal adjustments across targets" seems like an indecisive claim, since the need for target-specific tuning weakens the claim of a single, simple, and unified architecture.
>
> As stated in the manuscript:
>
> *“different multivector grades can be utilized for different properties without requiring task-specific readout layers or entirely new architectures. In this way, the combination of multivector embeddings and weighted geometric products enables the model to handle targets of varying geometric nature within a unified framework, requiring only minimal adjustments across targets."*
>
> Our intention with this phrasing was to emphasize that the representational framework and operations are unified, rather than to claim that a single architectural configuration performs optimally across all targets. We understand how this distinction may not have been entirely clear from the text, and we will make that distinction more clear in the revised manuscript.
> Specifically, multivector representations inherently comprise scalar, vector, bivector, and trivector grades, allowing the model to predict targets of different geometric types simply by extracting the relevant grade (e.g., scalar for scalar targets, vector for vector targets, etc.). Moreover, the geometric product itself can be viewed as a unifying operation, since it consistently mixes information across geometric levels in a way that is beneficial to targets of differing geometric nature.
>
> > W4: Concerns Regarding Computational Cost: As the author(s) acknowledge in the Discussion section, a practical downside of the proposed model is its "relatively high computational cost and parameter count." The main high-performing models presented are computationally intensive, which could be a barrier to adoption for larger-scale applications. Have authors figured out any way to optimize the already proposed lighter variants or any new ones?
>
> The computational complexity of GA-GNN is relatively high compared to older models such as PaiNN, but not when compared to the recent state-of-the-art models. In fact, it is comparable to GotenNet-B and has fewer parameters and a faster runtime than Equiformer v2 as shown in the table below:
>
> | Model           | Batch size | Avg. total epochs | Avg. time per epoch (s) | Avg. GPU days | Trainable parameters |
> |-----------------|------------|--------------------|---------------------------|---------------|------------------------|
> | Equiformer      | 128        | -                  | 425                       | 1.48          | 3.5M                   |
> | EquiformerV2    | 64         | -                  | 821                       | 2.85          | 11.2M                  |
> | EquiformerV2    | 48         | -                  | 847                       | 2.94          | 11.2M                  |
> | GotenNet-B      | 32         | -                  | 180                       | 1.15          | 9.2M                   |
> | GotenNet-S      | 32         | -                  | 117                       | 0.75          | 6.1M                   |
> | GA-GNN_alpha    | 100        | 508                | 169                       | 0.99          | 10.1M                  |
> | GA-GNN_mu       | 100        | 498                | 185                       | 1.06          | 10.2M                  |
> | GA-GNN_R2       | 100        | 1000               | 127                       | 1.46          | 2.8M                   |
> | GA-GNN_homo     | 100        | 321                | 171                       | 0.63          | 10.1M                  |
>
> These results are based on the reported complexity cost analysis in [1] and the runs for the four variants of GA-GNN used in our main evaluation. We will clarify these points in the discussion and include detailed runtime comparisons in the appendix in the final version. While we have not yet explored further optimizations of the lighter variants or designed new ones, the ablation results indicate that this would be a promising direction for future work.
>
> [1] Aykent, S. and Xia, T. (2025). *GotenNet: Rethinking Efficient 3D Equivariant Graph Neural Networks.*
> The Thirteenth International Conference on Learning Representations (ICLR).

---

### Meta-Review · Area_Chair_6oq4 · 2025-11-01

**Recommendation:** Accept (Poster)
**Confidence:** 4

**Metareview:**

The paper introduces an architecture based on geometric algebra for energy computation in molecular systems.

The paper is well organized, has clear exposition, and the experiments, while only on one dataset, include the comparison with various baselines.

It is known that computation using GA is slower than the standard Euclidean computation, but, as the author points out, other proposed architectures (Equiformer) could be more expensive.

While an additional criticism is that the proposed model does not have the best accuracy, the method shows competitive performance.

I am therefore inclined to accept the paper; this seems also to be the overall consensus among reviewers.

---

### Decision · Program_Chairs · 2025-11-05

**Decision:**

Accept (Spotlight)

**Comment:**

We recommend an oral and a poster presentation given the AC and reviewers recommendations.

A spotlight presentation refers to a poster selected for an oral highlight but not designated as a full oral presentation per the AC’s recommendation.